# Effect of PACAP on Heat Exposure

**DOI:** 10.3390/ijms24043992

**Published:** 2023-02-16

**Authors:** Keisuke Suzuki, Hiroki Yamaga, Hirokazu Ohtaki, Satoshi Hirako, Kazuyuki Miyamoto, Motoyasu Nakamura, Kaoru Yanagisawa, Takuya Shimada, Tomohiko Hosono, Hitoshi Hashimoto, Kazuho Honda, Kenji Dohi

**Affiliations:** 1Department of Emergency and Disaster Medicine, School of Medicine, Showa University, Tokyo 142-8555, Japan; 2Department of Anatomy, School of Medicine, Showa University, Tokyo 142-8555, Japan; 3Department of Functional Neurobiology, School of Pharmacy, Tokyo University of Pharmacy and Life Sciences, Tokyo 192-0392, Japan; 4Department of Health and Nutrition, University of Human Arts and Sciences, Saitama 339-0077, Japan; 5Center for Biotechnology, Showa University, Tokyo 142-8555, Japan; 6Laboratory of Molecular Neuropharmacology, Graduate School of Pharmaceutical Sciences, Osaka University, Osaka 565-0871, Japan; 7Molecular Research Center for Children’s Mental Development, United Graduate School of Child Development, Osaka University, Kanazawa University, Hamamatsu University School of Medicine, Chiba University, and University of Fukui, Osaka 565-0871, Japan; 8Division of Bioscience, Institute for Datability Science, Osaka University, Osaka 565-0871, Japan; 9Transdimensional Life Imaging Division, Institute for Open and Transdisciplinary Research Initiatives, Osaka University, Osaka 565-0871, Japan; 10Department of Molecular Pharmaceutical Sciences, Graduate School of Medicine, Osaka University, Osaka 565-0871, Japan

**Keywords:** heat stroke, PACAP, hypothalamus, core body temperature

## Abstract

Heat stroke is a life-threatening illness caused by exposure to high ambient temperatures and relative humidity. The incidence of heat stroke is expected to increase due to climate change. Although pituitary adenylate cyclase-activating polypeptide (PACAP) has been implicated in thermoregulation, the role of PACAP on heat stress remains unclear. PACAP knockout (KO) and wild-type ICR mice were subjected to heat exposure at an ambient temperature of 36 °C and relative humidity of 99% for 30–150 min. After heat exposure, the PACAP KO mice had a greater survival rate and maintained a lower body temperature than the wild-type mice. Moreover, the gene expression and immunoreaction of c-Fos in the ventromedially preoptic area of the hypothalamus, which is known to harbor temperature-sensitive neurons, were significantly lower in PACAP KO mice than those in wild-type mice. In addition, differences were observed in the brown adipose tissue, the primary site of heat production, between PACAP KO and wild-type mice. These results suggest that PACAP KO mice are resistant to heat exposure. The heat production mechanism differs between PACAP KO and wild-type mice.

## 1. Introduction

Heat-related illnesses, including heat stroke, are systemic diseases caused by exposure to high ambient temperature and relative humidity, and their incidence is estimated to increase owing to recent climate change [1,2]. Heat exposure is associated with damage to organs, including the liver, kidneys, gastrointestinal tract, heart, lungs, muscles, and central nervous system. Moreover, it can induce systemic inflammatory response syndrome, which may lead to multi-organ dysfunction or death [3,4]. However, the progression and worsening of life-threatening illnesses during heat exposure have not been studied in detail.

The human core body temperature is maintained at 36.5 °C [5]. Body temperature is regulated by balancing heat production and dissipation. Heat production is broadly classified into non-shivering heat thermogenesis, mainly found in brown adipose tissue (BAT), and shivering heat thermogenesis, which is produced mainly in skeletal muscles. In contrast, body temperature is dissipated by diffusion and evaporation from the skin, sweat, and breath. The main role of BAT is to protect the body from cold stress. Brown adipocytes in the BAT receive adrenergic signals via the β3-adrenergic receptor (b3AR, encoded by *Adrb3*) from the post-ganglionic neurons of the dorsomedial hypothalamus (DMH) and raphe pallidus (RPa) and increase free fatty acids and glucose through lipolysis by hormone-sensitive lipase (HSL, encoded by *Lipe*). Then, the cells produce heat from free fatty acids and glucose mediated by a key thermogenesis molecule, uncoupling protein 1 (UCP1) in the mitochondria [6,7].

Heat stroke mainly occurs in hot areas, as the higher ambient temperatures increase the core body temperature. However, in addition to heat, the high humidity of tropical regions including Japan in the summer prevents heat diffusion and vaporing from the body and critically increases the risk of heat stroke. Therefore, we developed an experimental mouse model (C57/BL6 strain) of heat stroke to resemble summer in tropical regions [8,9]. The model showed a biphasic abnormal thermoresponse, abnormal blood parameters, and multiple organ failure [8]. Moreover, the animals exhibited loss of motor coordination and cerebellar cells [9].

Pituitary adenylate cyclase-activating polypeptide (PACAP, encoded by *Adcyap1*) was isolated in 1989 from sheep hypothalamic extracts and belongs to the vasoactive intestinal polypeptide–secretin–glucagon peptide family [10,11]. PACAP is produced in many organs and plays diverse biological roles [12,13]. Several research groups have reported that endo- and exogenous PACAP prevents several types of organ injuries, including neuronal injuries [14,15,16,17], cardiomyopathy [18,19], hepatic damages [13,20,21], renal failure [13,22], and systemic inflammation [23]. Therefore, PACAP is suggested as a tissue protectant. PACAP has also been reported to be involved in thermoregulation. *Adcyap1*-deficient (PACAP knockout (KO)) mice exhibit reduced heat production during cold stress [24]. More recently, warm-sensitive neurons, which are located in the hypothalamus and are responsible for a homeostatic response to heat, have been identified and found to be molecularly defined by the co-expression of the neuropeptides, brain-derived neurotrophic factor and PACAP [25]. PACAP is also involved in the induction of hibernation and torpor-like states [26,27]. Although the role of PACAP in cold exposure has been demonstrated [24,28,29], no studies have reported the role of PACAP in heat exposure.

In the present study, we aimed to determine the role of endogenous PACAP on heat exposure using PACAP KO mice.

## 2. Results

### 2.1. Conditions of Heat Stroke on ICR Mice

We previously reported a heat stroke model in the C57/BL6 strain of mice [8,9], but not in the ICR strain of mice. In our preliminary study, when ICR mice were exposed to heat stress at approximately 41 °C ambient temperatures and >99% relative humidity for 60 min, the same as our previous conditions, these conditions were lethal to ICR mice. Therefore, we changed the conditions for the ICR mouse model in this study. In the present study, the incubator for heat stress was maintained at 36 °C ambient temperatures and >95% relative humidity. Figure 1a shows the actual ambient temperatures, relative humidity, and Wet Bulb Globe Temperature (WBGT) during the heat stress. WBGT is an environmental index that accounts for ambient temperatures and relative humidity in a room [30,31] and is recommended by the International Labour Organization to evaluate the environment of workers (https://www.ilo.org, accessed on 12 January 2023). The WBGT (ambient temperatures and relative humidity) before heat exposure was 37.3 ± 0.0 °C WBGT (34.3 ± 0.0 °C ambient temperatures and >99% relative humidity). This value was increased during heat exposure and was measured at 41.6 ± 0.2 °C WBGT (37.6 ± 0.2 °C ambient temperatures and >99% relative humidity) and 42.2 ± 0.0 °C WBGT (38.2 ± 0.0 °C ambient temperatures and >99% relative humidity) at 60 and 120 min, respectively.

### 2.2. Survival Rate during Heat Exposure

To determine the tolerance against heat exposure, 10 mice in wild-type and PACAP KO mice were subjected to heat exposure after 3 h of hydration and the survival rate was observed for 150 min (Figure 1b). The minimum number of mice possible was used in line with good ethical practice. The wild-type mice began to die after approximately 90 min of heat exposure, and the survival rate after 150 min was 20.0% (2/10). However, all the PACAP KO mice survived for more than 120 min, and the survival rate after 150 min was 80.0% (8/10, *p* < 0.01, by Kaplan–Meier survival analysis). Therefore, in the later study, the mice were subjected to heat exposure for 60 min.

### 2.3. Impact of Heat Exposure on Blood Count and Serum Biochemical Parameters

The blood count and serum biochemical parameters were measured 60 min after heat exposure. Changes in serum electrolytes and an increase in creatinine kinase (CK) are known to result from dehydration. Furthermore, in severe heat stroke patients and an experimental model mouse, a decreased platelet (Plt) count and multi-organ dysfunction in the liver and kidneys are often observed [8,32]. Several parameters exhibited significant differences between the two groups after 60 min of heat exposure (Table 1). The Plt count was decreased in the wild-type mice compared with the PACAP KO mice after 60 min of heat exposure (*p* < 0.01). Moreover, the levels of sodium ions (Na), total protein (TP), and albumin (Alb) in the wild-type mice were significantly greater than those in the PACAP KO mice, indicating dehydration. The aspartate aminotransferase (AST), lactate dehydrogenase (LDH), CK, blood urea nitrogen (BUN), and creatinine (Cre) levels in the wild-type mice were also significantly higher than those in the PACAP KO mice, suggesting multi-organ dysfunction in the liver, kidneys, and striated muscles. However, no significant differences were observed in any parameter between the two groups (*n* = 5 each group) before heat exposure.

### 2.4. Weight Loss before and after Heat Exposure

We previously reported that body weight (BW) decreased after heat exposure, likely due to dehydration [8]. In this study, we observed no significant differences in body weight before dehydration between the two groups. Three hours after water deprivation, both groups experienced a decrease in BW around 3%, but no significant difference was observed between the two groups. However, after 60 min of heat exposure, the decrease in BW of the wild-type mice (4.8 ± 0.2%) was significantly greater than that of the PACAP KO mice (2.7 ± 0.2%, *p* < 0.05) (Figure 2).

### 2.5. Morphological Observation in Liver and Kidney after 60 min of Heat Exposure

Multi-organ injuries are one of the major consequences of heat stroke [32], and we have reported such injuries in C57/BL6 mice [8]. Moreover, the serum biochemical parameters we observed in this study suggested that hepatic and renal injuries resulted from heat exposure. Therefore, we evaluated morphological changes after 60 min of heat stroke.

As shown in Figure 3a, uncertain hepatic sinuses around the central vein (zone 3 of the liver acinus) and some blood cells, such as granulocytes and lymphocytes, were observed in the representative image of the hepatic lobule in the wild-type mice. Moreover, the hepatocytes in zone 3 were vacuolated (Figure 3c), suggesting hepatic damage and ischemia. However, in the PACAP KO mice, the structure of hepatic sinuses was relatively maintained, and the hepatic vacuolations were less prominent than those in the wild-type mice (Figure 3b,d). We previously observed swelling and degeneration of tubular epithelial cells and urinary casts in the renal tissue of C57/BL6 mice 24 h after 1 h of heat exposure [8]. However, we observed no critical changes in the renal tissue of the wild-type or PACAP KO mice (Figure 3e,f) after 60 min of heat exposure.

### 2.6. Changes in Core Body Temperature in Wild-Type and PACAP KO Mice

Many studies on endo- and exogenous PACAP prevented several types of injuries, such as neuronal injuries, cardiomyopathy, and hepatic damage [12,13]. Therefore, we hypothesized that PACAP KO mice would have worse symptoms after heat exposure. However, in the present study, PACAP KO mice exhibited a greater survival rate and less intense symptoms after heat exposure. Dysregulated core body temperature is a hallmark of progression of symptoms after heat stroke, and recent studies have reported that PACAP might be a key molecule for thermohomeostasis [24,25].

We intraperitoneally implanted a small thermometer 2 weeks before heat exposure and determined the core body temperature in both groups of mice throughout the circadian rhythm and during heat exposure (Figure 4). The core body temperature increased temporally upon switching between light/dark cycles, although it was controlled between 35 and 38 °C. Upon comparing the circadian rhythm of the wild-type and PACAP KO mice, no significant differences were measured for 24 h before water deprivation (Figure 4a). Then, the core body temperature was compared during 180 min of dehydration, following 60 min of heat exposure (Figure 4b). The core body temperature in the wild-type mice was slightly increased 180 min before heat exposure, likely due to handling for BW measurement, and it gradually returned for 180 min dehydration. The core body temperature in the PACAP KO mice also increased handling for BW measurement and gradually returned to normal levels after 180 min dehydration. However, the decreases in the body temperature of the PACAP KO mice were slower than those of the wild-type mice, and the body temperatures of the PACAP KO mice were significantly different between −100 and −20 min, although insignificant at the onset of heat exposure. The core body temperature of the animals drastically increased during heat exposure and exceeded 40 °C within 10 min, and the changes in core body temperature were similar between the groups. However, the core body temperature of the wild-type mice was significantly higher at 20–30 min after heat exposure. The highest body temperatures were observed at 42 and 54 min after heat exposure in the wild-type (42.8 ± 0.11 °C) and PACAP KO (42.7 ± 0.18 °C) mice, respectively.

### 2.7. c-Fos Gene and Protein Expression in the Ventral Medial Preoptic Area (VMPO) of the Hypothalamus in Wild-Type and PACAP KO Mice after Heat Exposure

Increases in the core body temperature during heat exposure were less pronounced in the PACAP KO mice, suggesting that PACAP might contribute to thermoregulation. We then examined the gene and protein expression of c-Fos in the VMPO, which is a thermal center of the hypothalamus, in the wild-type and PACAP KO mice (Figure 5). Previously, it was reported that the *Adcyap1* is expressed in hypothermia [24,25].

Before heat exposure, 274 ± 35.1 c-Fos^+^ cells /mm^2^ were counted in the VMPO of the wild-type mice; however, the PACAP KO group had significantly fewer c-Fos^+^ cells at 103 ± 10.4 /mm^2^. The number of c-Fos^+^ cells was significantly increased after 60 min of heat exposure in the wild-type (534 ± 63.7 /mm^2^) and PACAP KO (458 ± 45.0 /mm^2^) mice. The number of c-Fos^+^ cells in the PACAP KO mice remained lower, though not significantly, than that in the wild-type mice (Figure 5a,b). *Fos* expression was also significantly lower in the PACAP KO mice than that in the wild-type mice before heat exposure. Similarly, after 30 min of heat exposure, *Fos* expression was significantly lower in the PACAP KO mice than in the wild-type mice, but not after 60 min of heat exposure (Figure 5c). To confirm the effects of PACAP in the VMPO, *Adcyap1* expression was also measured. The expression of *Adcyap1* was significantly increased during heat exposure in the wild-type mice (Figure 5d).

### 2.8. BAT Weight and Thermogenesis-Related Gene Expression in Wild-Type and PACAP KO Mice after Heat Exposure

BAT is a major target organ, which is downstream from the hypothalamus, for thermogenesis [5,7]. We examined the BAT anatomically (Figure 6) and observed the thermogenesis-related gene expression in the BAT (Figure 7) after and during heat exposure. The anatomical observation and wet weight of BAT in the wild-type and PACAP KO mice did not differ (Figure 6a,b).

We then examined thermogenesis-related gene expression in the wild-type and PACAP KO mice after heat exposure (Figure 7). No significant difference was observed between PACAP KO mice and wild-type mice in *Ucp1*, a key molecule for thermogenesis, under normal conditions (pre-heat exposure). However, heat exposure increased the expression of *Ucp1*. At 60 min after heat exposure, while the expression of *Ucp1* increased in the wild-type mice, it decreased to the normal level in PACAP KO mice (Figure 7a). The expression of *Adrb3* and *Lipe* in the PACAP KO mice was greater than that in the wild-type mice under normal conditions. Specifically, *Lipe* expression (*p* < 0.05) was significantly higher in PACAP KO mice than wild-type mice. During heat exposure, these levels decreased drastically and were sustained for 60 min in both groups; however, *Adrb3* expression (*p* < 0.05) was significantly higher in PACAP KO mice than in wild-type mice at 30 min after heat exposure (Figure 7b,c).

## 3. Discussion

PACAP has been known to play an important role in tissue protection [12,13,14,15], and we previously reported the neuro- and cardioprotective effects of exo- and endogenous PACAP [14,15,16,18]. Therefore, we initially suspected that PACAP KO mice would be vulnerable to heat exposure. However, in the present study, we determined that PACAP KO mice had a higher survival rate against heat exposure than wild-type mice and with fewer hematological and serum biochemical changes, including reduced evidence of hepatic damage. These results suggest that endogenous PACAP had a detrimental effect under heat stress.

Increased ambient temperatures also increase core body temperature, and abnormally high core body temperature causes multiple organ failure [3,4,33]. This is caused by hyperthermia, which is directly cytotoxic and affects membrane stability and transmembrane transport protein function [34]. Organ failure and cell death were found to be induced at a core body temperature above 41 °C [35,36]. In our previous study, we reported that the heat stroke model in C57/BL6 mice showed a biphasic abnormal thermoresponse [8]. Using a modified model for the present study, the core body temperature of the ICR mice finally exceeded 42 °C. From the above, this core body temperature was sufficient to cause multiple organ damage in mice. PACAP has been reported to be involved in thermoregulation [24]. Warm-sensitive neurons in the POA of the hypothalamus express PACAP, suppressing the non-shivering heat thermogenesis of BAT through the DMH [25,26,27]. On the other hand, intracerebral ventricle or intravenous injection of PACAP into PACAP KO mice increased body temperature within an hour [37]. Although PACAP is clearly involved in thermoregulation, it is not yet fully understood, especially how it functions during heat exposure.

In the present study, no difference in the core body temperature throughout the circadian rhythm was observed between wild-type and PACAP KO mice. Therefore, it is suggested that no significant differences in thermoregulation were present between them in normal non-stress conditions. However, the time for the core body temperature to reach 41 °C during heat exposure was significantly shorter in wild-type mice. Another report indicated hyperactivity in PACAP KO mice compared with wild-type mice, which may have affected the body temperature [38]. Additionally, in our study, the changes in core body temperature differed among the groups of mice. For example, upon handling the mice to weigh them 180 min before heat exposure, their core body temperature increased. While the core body temperature of the wild-type mice returned gradually to a homeostatic level, that of the PACAP KO mice decreased significantly more slowly. Moreover, the increase in core body temperature of PACAP KO mice during heat exposure was also slower than that of the wild-type mice, which quickly increased after heat exposure and reached 40 °C after 10 min. Therefore, we thought PACAP KO mice might have different thermoregulation under stress conditions.

Although the thermoregulation of neuronal networks has not been fully elucidated, preganglionic sympathetic nerves in the DMH and RPa project to postganglionic nerves, which innervate into the BAT. Norepinephrine is released from the nerve endings and activates HSL and UCP1 via b3AR in the BAT [39,40] (Figure 8a). The nerves from the VMPO area within the POA, which is considered to be the site of thermosensitive neurons [25], project to the preganglionic nerves and suppress the thermogenetic neuronal networks of BAT [41,42]. The gene expression and number of c-Fos^+^ cells in the present study were increased after heat exposure in the wild-type and PACAP KO mice. This result is similar to that previously reported in a rat model of heat stroke [43]. The *Adcyap1* levels also increased after heat exposure, suggesting that POA and PACAP contribute to heat stress. However, under normal conditions, the number of c-Fos^+^ cells and the gene expression of c-Fos were significantly lower in PACAP KO mice than those in wild-type mice. PACAP KO mice have been reported to have impaired thermogenesis of non-shivering heat in BAT during cold exposure because the noradrenaline contents in the BAT were lower in PACAP KO mice than in wild-type mice. Therefore, PACAP KO mice have been suggested to have a low survival rate in a low-temperature environment at the age of 2 weeks [24,44].

We did not note any differences in the core body temperature between both groups throughout the circadian rhythm or in the *Ucp1* levels in the BAT before heat exposure; however, the NA levels were lower in the PACAP KO mice [24,45]. In contrast, the expression of *Adrb3* and *Lipe* in the BAT was higher in the PACAP KO mice. Our results, as well as those of previous studies [24], suggest that PACAP KO balanced the thermogenesis network by enhancing *Adrb3* and *Lipe* expression in BAT as a compensatory response, although it exhibits lower nerve input from the hypothalamus in homeostatic conditions (Figure 8b, top panel). During heat exposure, the expression of *Adcyap1* in the POA of the wild-type mice increased and likely played a role in inhibiting the sympathetic nerve signals in the DMH and RPa to suppress the thermogenesis of BAT. However, simultaneously, the expression of *Adrb3* and *Lipe* also decreased in the BAT. Therefore, the thermoregulation in the POA, likely through PACAP, might have been counteracted because the core body temperature and *Ucp1* expression continued to decrease during heat exposure. However, the PACAP KO mice did not depend on PACAP and the POA for thermo-homeostasis and, instead, could depend on the suppression of NA [24] and enhanced *Adrb3* and *Lipe* expression. Therefore, downregulation of *Adrb3* and *Lipe* in the BAT during heat exposure might drastically influence thermogenesis, because the *Ucp1* expression decreased after 60 min of heat exposure (Figure 8b, bottom panel). Additionally, increased PACAP might contribute to increased core body temperature because intracerebral ventricle or intravenous injection of PACAP increased body temperature within an hour in previous studies [37,46,47]. Moreover, in previous studies, UCP1 has been found to be modulated by the nervous system and governed by the thyroid hormonal pathway [48], and in another study, PACAP was found to increase thyroid-stimulating hormone (TSH) in the pituitary gland [49]. Thus, PACAP might enhance thermogenesis through a different pathway because the expression of *Adcyap1* was increased in the hypothalamus of wild-type mice. Therefore, PACAP KO mice may experience suppressed thermogenesis, inhibiting heat-mediated tissue damage during heat exposure. Our study has some limitations. Firstly, we focused only on non-shivering heat production and did not consider other thermogenic factors, such as shivering heat and activity heat, including the influence of the adrenal gland. Moreover, we did not examine the effect of heat dissipation on maintaining thermo-homeostasis in detail. Secondly, we previously reported that heat exposure induces progressive multiple organ failure and cerebellar damage [8,9]. As PACAP is well known to protect various tissues from damage, in the present study, we only examined consequences immediately after heat exposure and did not examine the net effect of PACAP on the tissue damage following heat stress. Further studies are needed to demonstrate thermoregulation during heat stress and the effect of tissue damage on PACAP. Thirdly, the body’s response to heat exposure may be different in humans and mice due to differences in sweating functions. For clinical application, for example, it is necessary to investigate PACAP in human serum in heat exposure.

## 4. Materials and Methods

### 4.1. Animals

Male ICR mice (8–15 weeks old) were used in this study. PACAP KO mice have been described previously [38]. The animals used in the present study were obtained by breeding between PACAP heterozygous KO pair mice under specific pathogen-free conditions in the animal facility of Showa University. The mice underwent a topped tail to determine the genotyping and ear punch for identification at 4 weeks after birth. Then, the tails were digested with Proteinase K (Nakalai Tesque, Tokyo, Japan) to isolate the genomic DNA with a tissue DNA isolation kit (Kurabo, Tokyo, Japan) supported by a Nucleic Acid Extraction System (NA-2000, Kurabo). Then, the aliquots were amplified using the specific primers [38], and the PCR products were electrophoresed with 2% agarose and detected with ChemiDoc XRS Plus (Bio-Rad, Hercules, CA, USA) after staining by ethidium bromide (Sigma, St. Louis, MO, USA). Representative PCR signals are shown in Figure 9. The mice were allowed free access to food and water and were maintained in a 12 h light/dark cycle at room temperature (24 ± 2 °C) with constant humidity (40 ± 15%). All experimental procedures involving animals and clinical data were approved and monitored by the Institutional Animal Care and Use Committee of Showa University (#02042, 03030, and 04062), which adhered to the ARRIVE guidelines. All methods were performed in accordance with the relevant guidelines and regulations.

### 4.2. Heat Stroke Model

A mouse model of heat stroke that mimicked temperate subtropical weather conditions using WBGT as an indicator has been reported previously [8]. We followed these experiments with minor modifications. Briefly, the semi-enclosed heat stroke chamber (200 mm × 340 mm × 300 mm), which was custom ordered (Niigata Co Ltd., Yokohama, Japan), was made from acrylic materials like a greenhouse. A digital thermo-hygrometer (AD-5696, A&D Company, Tokyo, Japan) was placed inside to monitor the ambient temperatures, relative humidity, and WBGT. An emission tube of an ultrasonic humidifier (SMB-1, Sakura, Tokyo, Japan) was connected to the upper side wall of the chamber for humidification (Figure 10). Then, the heat stroke chamber was placed inside an incubator (Program Incubator IN604W, Yamato Scientific, Tokyo, Japan) and pre-heated at 36 °C for 180 min. The humidifier was initiated 60 min before heat exposure to create a high-WBGT environment.

Meanwhile, the water intake of the mice was restricted to induce mild dehydration, and the mice were quickly placed in the heat stroke chamber. Then, the animals were subjected to heat exposure for 30–150 min depending on the purpose of the experiments described below. After heat exposure, blood and tissue samples were immediately (within 20 min) obtained from the mice, as described below. Ten mice (usually five wild-type and five PACAP KO mice) were subjected to heat exposure in each experiment. The experiment employing 60 min of heat exposure was repeated six times (30 wild-type and 30 PACAP KO mice), and 13 mice (7 wild-type and 6 PACAP KO mice) died as a result.

### 4.3. Examination of Survival Rate

To examine their vulnerability to heat exposure, wild-type (*n* = 10) and PACAP KO (*n* = 10) mice were subjected to heat exposure for 150 min, and the survival was observed by monitoring their breathing. The time of death was defined as 5 min after the appearance of akinesia and apnea. These studies were repeated twice to confirm reproducibility, and 10 mice in each group were examined.

### 4.4. Preparation of Histological Samples and Morphological Observation

Immediately after 60 min of heat exposure, the mice were anesthetized with an overdose of sodium pentobarbital (100 mg/kg, i.p.). Blood samples were then collected from the right ventricle of the heart. The mice were then perfused transcardially with 0.9% NaCl followed by 4% paraformaldehyde (PFA), and the brain, interscapular BAT, liver, kidneys, and intestines were removed. The livers and kidneys were postfixed in 10% neutralized formalin for preparing paraffin-embedded sections (4 μm thick), and HE staining was conducted. After weighing the BAT, the BAT and brain samples were postfixed in 4% PFA for preparing frozen sections, as described below.

An aliquot of blood (approximately 100 μL) was collected in a sample tube containing EDTA-2Na (Capiject CJ-NA, Terumo, Tokyo, Japan) and kept on ice to measure the complete blood count (CBC). Other blood samples were collected, centrifuged at 1500× *g* for 10 min to collect the serum, and frozen until use to measure the biochemical parameters.

### 4.5. CBC and Biochemical Parameters in Serum

The CBC was measured in the EDTA-blood samples using VetScan HM5 (ABAXIS, Union City, CA, USA). In the serum samples, we measured the electrolytes (Na, K, and Cl), TP, ALB, total bilirubin (T-bil), AST, ALT, ALP, LDH, CK, BUN, Cre, and TSH. Electrolytes were assayed by the ion-selective electrode method; TP was assayed using the Biuret method; ALB was assayed with the BCG method; BUN was assayed by the urease-GLDH method; T-bil, Cre, and LA were assayed via the enzymatic method; AST, ALT, ALP, LDH, and CK were assayed with the MDH-UV method; and TSH was assayed using the Luminex method. All assays were performed using HITACHI 7180 (Tokyo, Japan).

### 4.6. Measurement of Core Body Temperature

To monitor the core body temperature throughout the circadian rhythm and during the 60 min heat exposure, the animals were implanted with a small thermometer (Thermochrone type G, KN laboratories, Osaka, Japan) in their abdominal cavity. Briefly, the mice (five each wild-type and PACAP KO) were anesthetized with 4% sevoflurane (Wako, Osaka, Japan) through N_2_O/O_2_ (70/30%) inhalation, and an incision was made (approximately 1.0 cm) on the midline of the abdomen under aseptic conditions. The thermometer was then implanted between the abdominal aorta and intestinal membranes, and the incision was closed using sutures. The animals were maintained for 2 weeks for recovery, until they were subjected to heat exposure for 60 min. When the animals were euthanized, the thermometer was removed, and the core body temperature was recorded. The core body temperature was recorded every 3 min during each experimental period after implantation.

### 4.7. Immunohistochemistry and Cell Count of Hypothalamus

After the perfused fixation, the brain and BAT samples were incubated in 20% sucrose in 0.1 M phosphate buffer (PB) at a pH of 7.2 for two nights and embedded in liquid-nitrogen-cooled isopentane using an embedding solution (20% sucrose in 0.1 M PB: O.C.T. compound (Sakura Finetech, Tokyo, Japan) = 2:1).

Coronal brain samples (0.3–0.5 mm from the bregma) were sectioned into 25 μm slices using a cryostat (Hyrax50, Carl Zeiss, Inc.; Oberkochen, Germany) and transferred into a cryoprotectant solution (0.1 M PB, 30% sucrose, 1% polyvinylpyrrolidone, 30% ethylene glycol [50]) in a multi-well tissue culture plate at –30 °C.

The 25 μm floating sections were washed with PBS containing 0.1% Triton X-100 (PBST) and incubated with 30% formic acid for 5 min to retrieve the antigens. After the application of 0.3% H_2_O_2_ to quench the internal peroxidase reactions, the sections were incubated in 5% normal horse serum (Vector, Burlingame, CA, USA) in PBST to block non-specific reactions for 1 h. Subsequently, the sections were incubated overnight with polyclonal rabbit anti-c-Fos (1:5000; Santa Cruz Biotech, Santa Cruz, CA, USA). After washing with PBST, the sections were incubated with biotinylated goat anti-rabbit immunoglobulin G (1:1000, Invitrogen, CA, USA) for 2 h. They were then incubated in an avidin–biotin complex solution (Vector) followed by DAB as a chromogen. The sections were mounted on coating slide glass (0.01% poly-L-lysine-coated glass slide (Sigma, St. Louis, MO, USA)) and enclosed by Malinol (Muto Pure Chemicals, Tokyo, Japan). The sections were observed using a microscope (Olympus BX53; Olympus, Tokyo, Japan), and images were taken with Olympus cellSens standard 2.1 (Olympus). These staining experiments were carried out on five wild-type and five PACAP KO mice.

The number of c-Fos^+^ cells in the VMPO of the hypothalamus was manually counted with the aid of the process mode in Olympus cellSens standard 2.1. Three of the five sections at 0.3–0.5 mm from the bregma in each mouse were randomly selected, and the number of c-Fos^+^ cells was counted. The area of the VMPO was traced manually (approximately 0.3 mm^2^), and the cell counts were calculated per mm^2^.

### 4.8. Polymerase Chain Reaction (PCR)

The BAT was removed from the other set of mice (10 wild-type and 10 PACAP KO) at 0, 30, and 60 min after heat exposure. The brain was trimmed at the position between 1.0 and 0 mm coronal from the bregma, and the ventromedial area from the anterior part of anterior commissures, including the VMPO of the hypothalamus, was dissected. The BAT and brains were snap-frozen in liquid nitrogen and stored at −80 °C until use. Total RNA was extracted from the tissues of each mouse using Trizol (Invitrogen) in accordance with the manufacturer’s protocol and converted into cDNA with ReverTra Ace^®^ qPCR RT Master Mix for BAT (TOYOBO, Osaka, Japan) or with a High-Capacity RNA-to-cDNA Kit for brain tissue (Applied Biosystems, Foster City, CA, USA).

Gene expression was quantified with SYBR-green-based quantitative real-time PCR with specific primers. The cDNA of BAT was analyzed on the Mic qPCR Cycler (Bio molecular systems, Upper Coomera, QLD, Australia) and the Luna Universal qPCR Master Mix (New England Biolabs, Ipswich, MA, USA). The cycling conditions for assays consisted of 1 min at 95 °C, followed by 45 cycles of 15 s at 95 °C and 30 s at 60 °C.

The cDNA of the brain tissue was analyzed on an Applied Biosystems 7900HT Fast Real-Time PCR Systems (Applied Biosystems, Lincoln, CA, USA), using the SYBR Premix Ex TaqTMIIreagent (TaKaRa, Shiga, Japan). The cycling conditions for assays consisted of 30 s at 95 °C, followed by 45 cycles of 5 s at 95 °C and 31 s at 61 °C.

Relative gene expression was calculated using the absolute quantification method. The gene expression of *Ucp1*, *Lipe*, and *Adrb3* was determined using their specific primers. *Gapdh* was used as a housekeeping gene to normalize the cDNA levels. All specific primers are presented in Table 2. All data are expressed as the relative level (fold) to compare with the value at 0 min (as a pre-heat-exposure control) after normalization against mouse *Gapdh* (*n* = 7–12 mice at each time point).

The gene expression of *Adcyap1* and *Fos* was determined using their specific primers. *Actb* was used as a housekeeping gene to normalize the cDNA levels. All specific primers are presented in Table 2. All data are expressed as the relative level (fold) to compare with the value at 0 min (as pre-heat-exposure control) after normalization against mouse *Actb* (*n* = 7–12 mice at each time point).

### 4.9. Statistical Analysis

Data were expressed as the mean ± standard error of the mean. We constructed survival curves using the Kaplan–Meier method. Statistical comparisons between the wild-type and PACAP KO mice were made using Student’s *t*-test. A *p*-value < 0.05 was considered to indicate statistical significance. Analyses were performed using Statcel4 (OMS Ltd., Tokyo, Japan).

## 5. Conclusions

In conclusion, we demonstrated that PACAP KO mice of ICR mice exhibited a better survival rate and improved clinical symptoms compared with wild-type mice after heat exposure. The PACAP KO mice demonstrated slower responses to increase core body temperature during heat exposure, suggesting that they are resistant to heat exposure. The thermoregulating gene expression in the POA of the hypothalamus and BAT also differed between the two groups of mice. Our results provide new knowledge on thermoregulation of heat stress via PACAP in the POA-BAT axis. Furthermore, these findings may lead to the establishment of new treatment and prevention methods to control heat production in heat stroke.

## Figures and Tables

**Figure 1 ijms-24-03992-f001:**
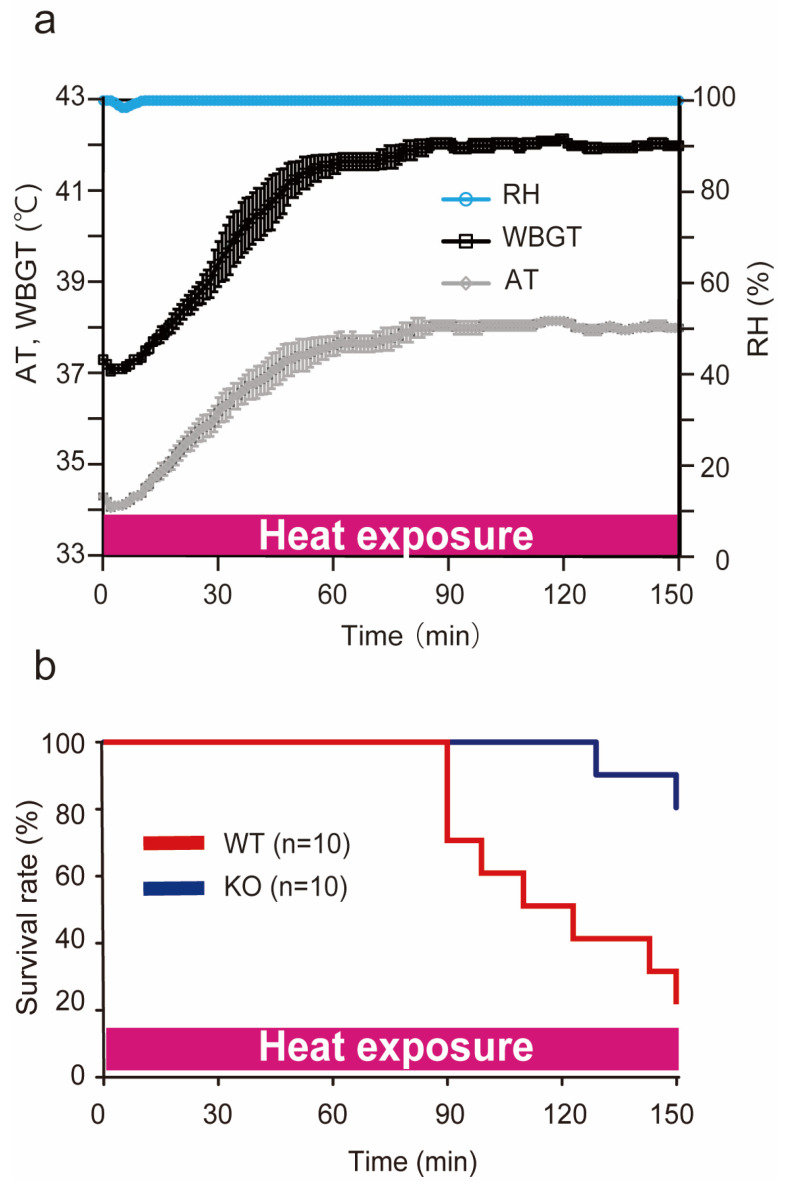
Conditions of heat stroke in ICR mice and the survival rate during heat exposure. (**a**) During heat exposure, WBGT (°C), ambient temperatures (ATs) (°C), and relative humidity (RH) (%) were measured with a digital thermo-hygrometer. The measurements were repeated four times and expressed as the mean ± SD. The WBGT (AT and RH) before heat exposure was 37.3 ± 0.0 °C (34.3 ± 0.0 °C and 99.9 ± 0.0%). It reached 42.0 ± 0.1 °C (38.0 ± 0.0 °C and 99.9 ± 0.0%) at 150 min. (**b**) The survival rate (%) of the wild-type (WT) and PACAP KO (KO) mice under heat exposure. WT (red line) and PACAP KO (blue line) mice (*n* = 10 each) were exposed to heat exposure for 150 min, and the survival time was observed. The survival rates at 150 min were 20.0% and 80.0% in the WT and PACAP KO groups, respectively (*p* < 0.05, by Kaplan–Meier survival analysis).

**Figure 2 ijms-24-03992-f002:**
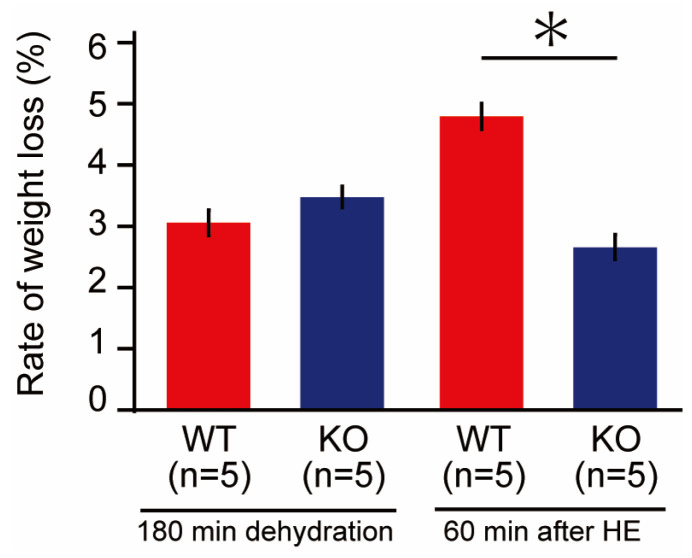
Decrease in body weight (BW; %) of the wild-type and PACAP KO mice. After 3 h water restriction, BW decreased by approximately 3% in each group. The decrease in BW after 60 min of heat exposure was significantly higher in the wild-type mice than in the PACAP KO mice (Student’s *t*-test, * *p* < 0.05).

**Figure 3 ijms-24-03992-f003:**
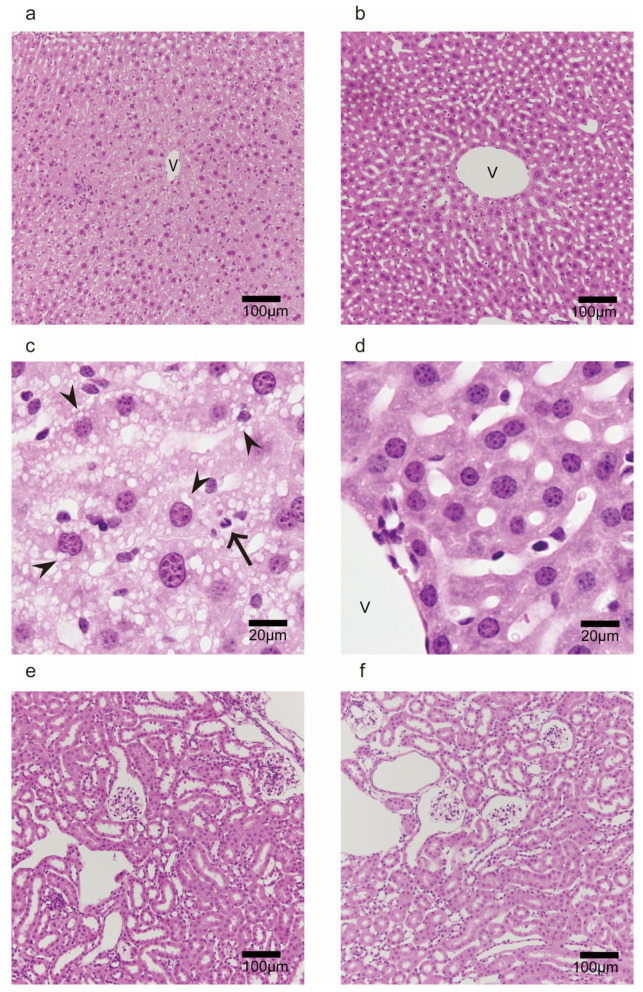
Morphological observations (HE staining) of livers (**a**–**d**) and kidneys (**e**,**f**) of the wild-type (**a**,**c**,**e**) and PACAP KO (**b**,**d**,**f**) mice after 60 min of heat exposure. (**a**) The hepatic sinuses surrounding the area of the central vein (V) were uncertain. (**c**) The higher-magnification image demonstrated that the vacuolations in the hepatocytes (arrowhead) and leukocytes (arrow) were present in the wild-type mice and relatively suppressed in the PACAP KO mice (**b**,**d**). The renal tissues did not show obvious pathological changes or differences between the groups (**e**,**f**).

**Figure 4 ijms-24-03992-f004:**
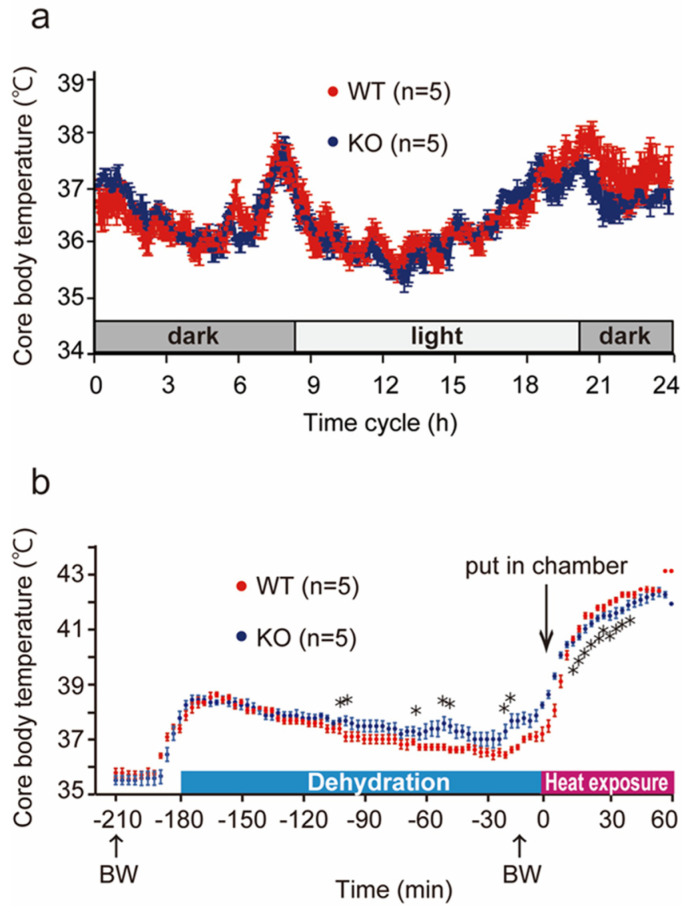
Changes in core body temperature throughout the circadian rhythm and heat exposure. (**a**) The core body temperature in the WT (*n* = 5) and PACAP KO (KO; *n* = 5) mice during the circadian rhythm. The core body temperature changed between 35 and 38 °C in a day. However, no significant differences were observed between the groups. (**b**) The core body temperature in the wild-type and PACAP KO mice during 180 min of dehydration and 60 min of heat exposure. The core body temperature slightly increased after measuring BW (−180 min) and gradually returned to physiological levels. The core body temperature drastically increased in the heat stroke chamber and exceeded 42 °C. The PACAP KO mice exhibited a similar pattern to that of the wild-type mice. However, the PACAP KO mice exhibited a significantly slower decrease during dehydration and increase during heat exposure. Data are expressed as the mean ± SE. * *p* < 0.05 (Student’s *t*-test).

**Figure 5 ijms-24-03992-f005:**
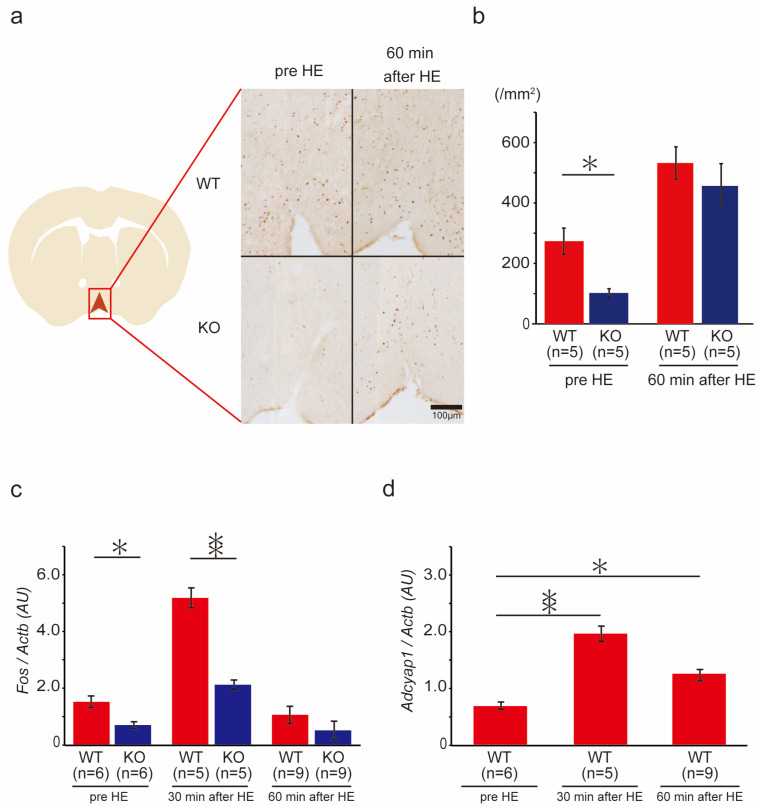
c-Fos immunoreaction and gene expression in the VMPO of the hypothalamus in the wild-type and PACAP KO mice after heat exposure (HE). (**a**) Immunostaining of c-Fos in the VMPO in the wild-type and PACAP KO mice. Schematic drawing from Paxinos and Flanklin’s atlas and representative photomicrographs of coronal sections from the VMPO at 0.3–0.5 mm from the bregma [32]. (**b**) Quantitative analysis of c-Fos+ cell count in the VMPO (*n* = 5 of each group). The PACAP KO mice exhibited significantly fewer c-Fos+ cells than the wild-type mice pre-HE (*p* < 0.05). The number significantly increased in both groups after HE, with no significant difference. (**c**) c-Fos expression before and after HE. c-Fos expression was significantly lower in PACAP KO mice than in wild-type mice before HE. (**d**) *Adcyap1* was expressed in the VMPO of the wild-type mice and increased after HE. Data are expressed as the mean ± SE. * *p* < 0.05, ** *p* < 0.01 (Student’s *t*-test).

**Figure 6 ijms-24-03992-f006:**
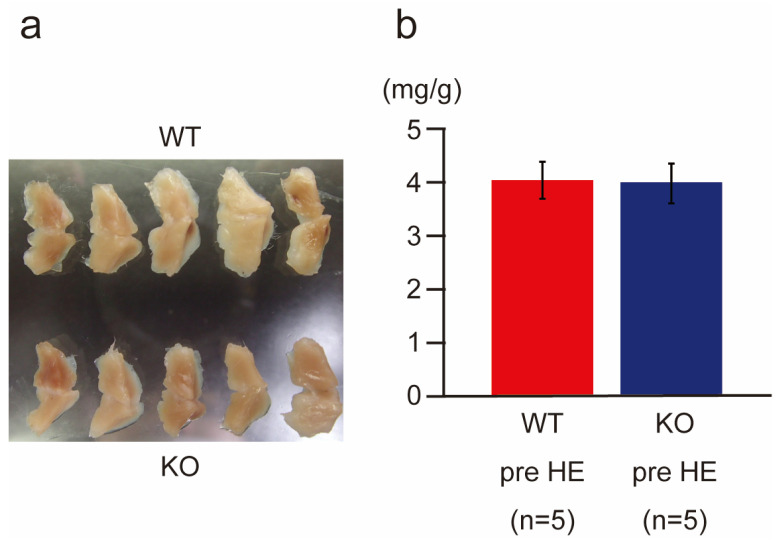
Morphohistological comparison of brown adipose tissues (BATs) in the wild-type and PACAP KO mice. (**a**) Gross comparison of BAT between the wild-type (*n* = 5) and PACAP KO (*n* = 5) mice. No significant difference was observed between the two groups. (**b**) Wet weight of BAT also demonstrated no significant differences between the two groups. Data are shown as the mean ± standard error (SE).

**Figure 7 ijms-24-03992-f007:**
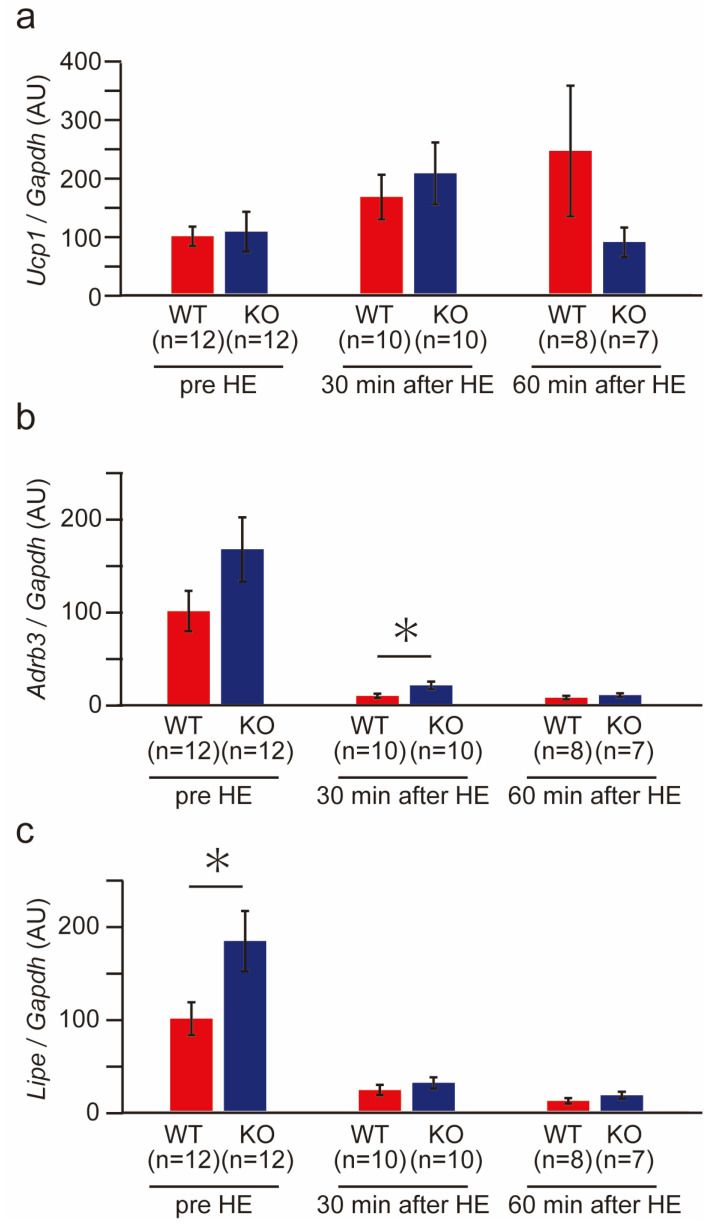
Thermogenesis-related gene expression of BAT in the wild-type and PACAP KO mice before and after heat exposure. The expression of thermogenesis-related genes, such as *Ucp1* (**a**), *Adrb3* (**b**), and *Lipe* (**c**), was determined with SYBR green-based real-time PCR before and 30 and 60 min after heat exposure in the wild-type (*n* = 8–12) and PACAP KO (*n* = 7–12) mice. The gene expression was normalized with a housekeeping gene, *Gapdh,* and expressed in arbitrary units (AUs). All data are shown as the mean ± standard error (SE). * *p* < 0.05 (Student’s *t*-test).

**Figure 8 ijms-24-03992-f008:**
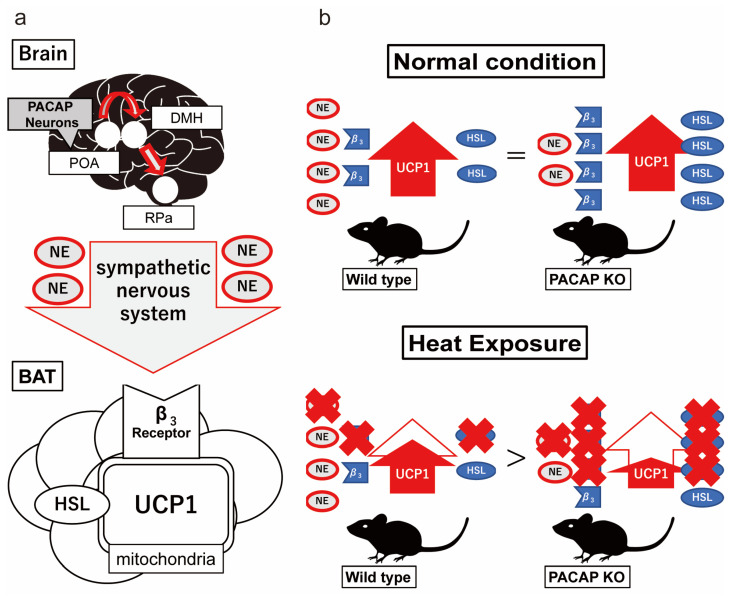
Schematic pathway of non-shivering heat thermogenesis in BAT and putative thermoregulation of PACAP under normal and heat conditions. (**a**) Schematic nerve pathway for non-shivering thermoregulation with BAT. Non-shivering thermogenesis is regulated by the sympathetic nerve innervation. Preganglionic sympathetic nerves from the DMH and RPa project to postganglionic nerves, which innervate into the BAT. The nerves release noradrenaline (NA) and enhance a key thermogenetic molecule, UCP1, mediated by β3-adrenaline receptor (b3AR) and hormone-sensitive lipase (HSL) in the BAT. (**b**) Putative thermoregulation of the wild-type and PACAP KO mice under normal and heat conditions. The NA contents in the BAT were lower in PACAP KO mice than wild-type mice [24,45]. To compensate for their lower NA contents, the PACAP KO mice exhibited increased expression of b3AR and HSL, therefore, maintaining the UCP1 level and core body temperature. Simultaneously, heat exposure increased PACAP in the POA to inhibit the DMH and RPa nerve signals and decreased the expression of b3AR and HSL to suppress thermogenesis. On the other hand, because of the POA signaling, and because downregulation of b3AR and HSL was involved in the same pathway, the thermoregulation was less effective. However, in PACAP KO mice, thermogenesis depends highly on b3AR and HSL in normal conditions. Therefore, strong suppression of b3AR and HSL expression might result in the suppression of UCP1 and thermogenesis.

**Figure 9 ijms-24-03992-f009:**
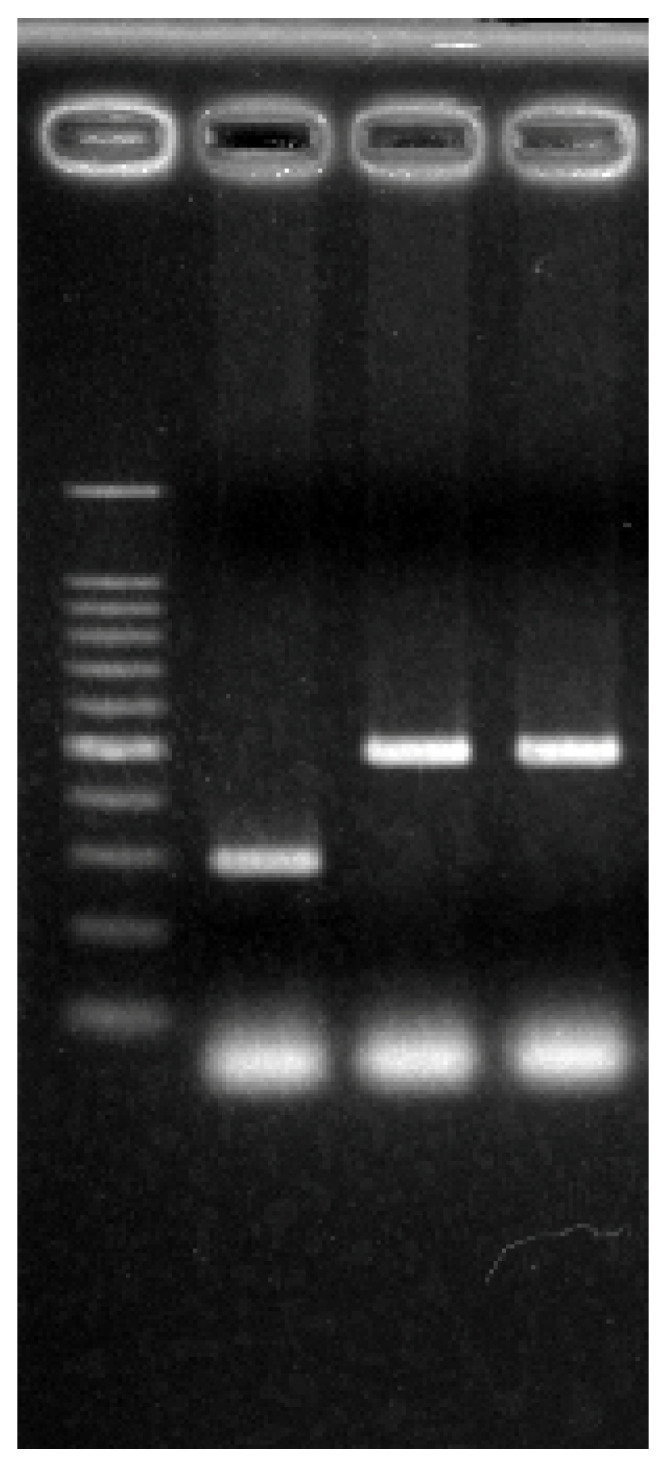
Representative images of the PACAP KO and wild-type mice. Higher bands indicate homozygous and lower bands indicate wild type.

**Figure 10 ijms-24-03992-f010:**
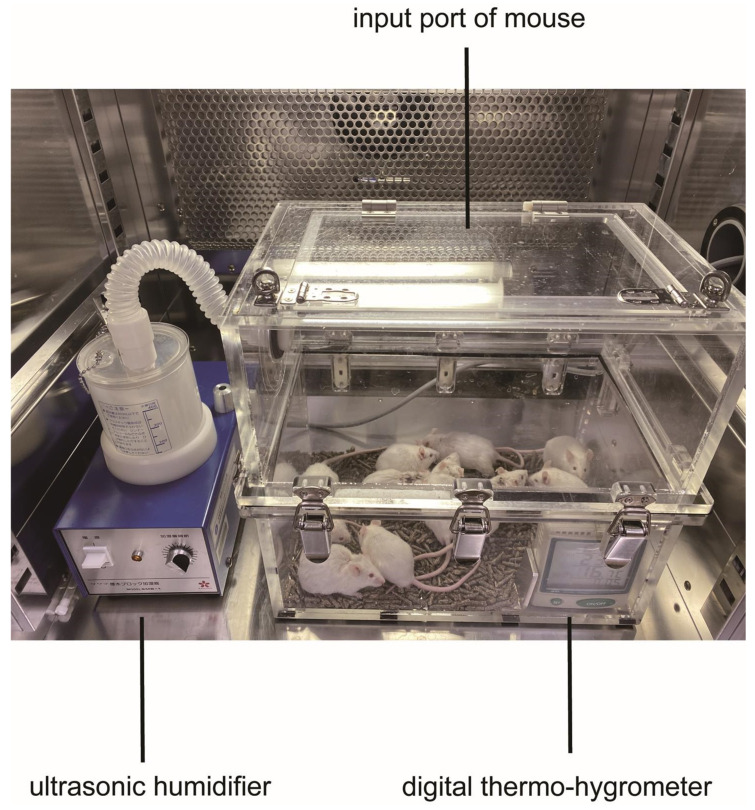
Heat stroke chamber: Ten mice were placed in an acrylic box from the top input port. Humidity was controlled by an ultrasonic humidifier and recorded by a digital thermo-hygrometer.

**Table 1 ijms-24-03992-t001:** Blood count and serum biochemical parameters before and after 60 min of heat exposure.

	pre HE			60 min after HE		
	WT (n = 5)	KO (n = 5)	*p*	WT (n = 5)	KO (n = 5)	*p*
WBC × 10^6^ (/L)	3.0 ± 1.4	1.8 ± 0.6		3.3 ± 0.8	5.3 ± 3.3	
RBC × 10^12^ (/L)	10.2 ± 0.3	9.8 ± 0.2		11.4 ± 0.7	10.7 ± 0.1	
Hb (g/dL)	15.0 ± 0.6	14.8 ± 0.4		20.0 ± 1.7	17.9 ± 0.3	
Hct (%)	44.6 ± 1.1	44.5 ± 0.6		50.5 ± 2.9	46.1 ± 0.5	
Plt × 10^9^ (/L)	622.0 ± 39.2	591.0 ± 27.4		176.0 ± 43.0	491.0 ± 29.3	**
Na (mmol/L)	151.0 ± 0.9	153.0 ± 0.9		156.0 ± 2.2	151.0 ± 0.7	**
K (mmol/L)	6.6 ± 0.3	5.7 ± 0.8		6.4 ± 1.0	7.6 ± 0.4	
Cl (mmol/L)	105.0 ± 1.0	106.0 ± 0.7		106.0 ± 2.2	112.0 ± 1.6	
TP (g/dL)	4.7 ± 0.1	4.8 ± 0.1		5.3 ± 0.2	4.6 ± 0.1	**
Alb (g/dL)	2.6 ± 0.0	3.0 ± 0.1		3.4 ± 0.3	2.7 ± 0.1	*
AST (IU/L)	68.0 ± 8.5	60.0 ± 6.0		390.0 ± 96.5	143.0 ± 17.9	**
ALT (IU/L)	34.0 ± 7.9	30.0 ± 5.8		172.0 ± 37.6	72.0 ± 22.1	
ALP (IU/L)	141.0 ± 14.2	212.0 ± 22.2		216.0 ± 23.2	239.0 ± 21.5	
LDH (IU/L)	323.0 ± 310.9	921.0 ± 91.2		3480.0 ± 1011.3	1057.0 ± 74.3	*
CK (IU/L)	508.0 ± 147.4	747.0 ± 103.7		1515.0 ± 134.5	1088.0 ± 165.1	*
BUN (mg/dL)	27.6 ± 1.1	26.5 ± 1.5		73.6 ± 5.2	30.7 ± 0.8	**
Cre (mg/dL)	0.1 ± 0.0	0.2 ± 0.0		0.4 ± 0.1	0.2 ± 0.0	**

WBC, white blood cell; RBC, red blood cell; Hb, hemoglobin; Hct, hematocrit; Plt, platelet; Na, sodium; K, potassium; Cl, chlorine; TP, total protein; Alb, albumin; T-bil, total bilirubin; AST, aspartate aminotransferase; ALT, alanine aminotransferase; ALP, alkaline phosphatase; LDH, lactate dehydrogenase; CK, creatinine kinase; BUN, blood urea nitrogen; Cre, creatinine. All data are expressed as the mean ± SE. * *p* < 0.05, ** *p* < 0.01 (Student’s *t*-test).

**Table 2 ijms-24-03992-t002:** Primer used in the present study.

Name	Sequence		Size
*Adcyap1*	Forward	AACCCGCTGCAAGACTTCTATGAC	125
	Reverse	TTAAGGATTTCGTGGGCGACA	
*Fos*	Forward	CGGGTTTCAACGCCGACTA	165
	Reverse	TTGGCACTAGAGACGGACAGA	
*Actb*	Forward	GGCTGTATTCCCCTCCATCG	154
	Reverse	CCAGTTGGTAACAATGCCATGT	
*Ucp1*	Forward	GTCAACAGCAAAAGCCACAA	206
	Reverse	TCTGGGGTCAGAGGAAGAGA	
*Lipe*	Forward	GGCTCACAGTTACCATCTCACC	106
	Reverse	GAGTACCTTGCTGTCCTGTCC	
*Adrb3*	Forward	CAGCCAGCCCTGTTGA	60
	Reverse	CCTTCATAGCCATCAA	
*Gapdh*	Forward	TGTGTCCGTCGTGGATCTGA	57
	Reverse	CCTGCTTCACCACCTTCTTGAT	

## Data Availability

All relevant data within the paper are available from the corresponding author upon reasonable request.

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
