# Peer review of "Effect of PACAP on Heat Exposure"

_ijms, 2023, doi:10.3390/ijms24043992_

Round 1

Reviewer 1 Report

Thank you for submitting this detailed, and well prepared manuscript detailing new knowledge relating to the effect of PACAP on mice exposed to heat stress. Your attention to detail in the manuscript presentation is to be commended.

There is one point that perhaps needs to be clarified in relation to the strain of mice used in the study. Line 87, you explain that a previous mouse heatstroke model used C57/BL6 mice, but the ICR strain appeared more susceptible to heat stress as they all died using this experimental procedure. When you refer to your “preliminary study” on line 88, do you mean a preliminary study that was part of THIS manuscript, or are you referring to a preliminary study that was part of Reference 6? You do not mention using C57/BL6 mice in the methods, so I am assuming the later? However, if you did perform ICR versus C57/BL6 comparisons this should be added to the methods.

Related to this, given the results of this preliminary study, your discussion should acknowledge that whilst PACAP KO mice appear to manage heat exposure better than the wild type, the ICR mice in general appear to have an inherent sensitivity to heat in comparison to the C57/BL6 mice. Perhaps you could hypothesis a reason for this difference?

Personally, I found the use of acronyms a little confusing at times as there are many. Consider writing out the less common ones in full for readability? E.g. ambient temperature instead of AT?

In general, review where the term “heat stroke” has been used in lieu of “exposure to heat stress”. Whilst this is a heat stroke model, heat stroke (or severe heat-related illness) is itself a diagnosis, so your mice were exposed to heat stress during the experiment, which would be the more accurate description of the conditions.

Some specific points to consider:

Line 38 – there has been a shift towards using the term “heat-related illness” instead of heat stroke within some medical field (See Yamamoto et al, 2015 and 2018 - http://www.mdpi.com/1660-4601/12/9/11770 and http://www.mdpi.com/1660-4601/15/9/1962), recognising that heat stroke is the most severe form of a continuous spectrum of heat induced illnesses. Consider re-phrasing?

Line 45 – please reference this temperature. For example Hausmann et al 2018 (http://link.springer.com/10.1007/s11606-018-4610-8) recently reviewed human normal temperatures and proposed different values.

Line 49 – should “diffusing and vapouring” be “diffusion and evapouration”?  

Line 70 – remove “basically”

Line 80 – this may be a journal specific presentation style, but results are not normally presenting alongside the aims? Please check.

Line 88 – as noted above, please clarify if this preliminary study was part of this research project or the cited work? “were weak against heat stress” consider rephasing to “were more susceptible to heat stress”.

Figure 1 – ensure legend is below figure.

Line 111 – please specify the groups. Also, did you perform any sample size estimations to determine the group size? Given the welfare impact on the mice used, it would be good to know that the minimum number possible were used in line with good ethical practice.

Line 153 – please rephase “symptoms”, perhaps consequences? Symptoms are subjective, reported experiences of human patients so the term is not appropriate for an animal model.

Line 187 – “insignificant” consider changing to no significant differences.

Lines 193-200 – some phrasing in this section is unclear, e.g. “they were significant between -100 and -200”, do you mean there were significant differences in temperature, or in the rate of temperature change? Please clarify.

Line 213 – consider rephrasing “suppressed”, as this implies there is a mechanisms actively suppressing temperature increase, whereas you appear to report that instead the KO mice instead were protected from mechanisms that contributed to temperature increase? So perhaps “increases in the cT during exposure to heat stress were less pronounced in the KO mice”?  

Line 280 – “were better blood count” consider instead “had a higher survival rate against heat exposure than wild-type mice with fewer haematological and serum biochemical changes, including reduced evidence of hepatic damage”.

Line 281 – “deteriorating role under heat stress” consider instead “had a detrimental effect under heat stress”.

Lines 283 – 292 – consider expanding this paragraph to clarify the links between the main statements, at the moment the shorter standalone statements do not appear to be related?

Line 295 – consider rephrasing: “no significant differences in thermoregulation were present in normal non-stress…”

Line 299 – “in this study” do you mean in the present study, or do you mean in study reference 36? Please clarify.

Line 305 – please remove bold

Line 520 – please clarify that the wild type mice were ICR mice, at the moment this statement is misleading, as this heatstroke model used a lower ambient temp compared to the previous C57/BL6 mice.

Line 542 – “not applicable” is not appropriate, either the data should be made available in an open access formal, or this section to detail how interested parties can obtain the data, e.g. who to contact.

Author Response

Response to Reviewer #1

Thank you very much for providing positive and important comments to improve our manuscript. We read your comments repeatedly and revised our manuscript accordingly. We also responded to the reviewer’s comments as follows:

  1. There is one point that perhaps needs to be clarified in relation to the strain of mice used in the study. Line 87, you explain that a previous mouse heatstroke model used C57/BL6 mice, but the ICR strain appeared more susceptible to heat stress as they all died using this experimental procedure. When you refer to your “preliminary study” on line 88, do you mean a preliminary study that was part of THIS manuscript, or are you referring to a preliminary study that was part of Reference 6? You do not mention using C57/BL6 mice in the methods, so I am assuming the later? However, if you did perform ICR versus C57/BL6 comparisons this should be added to the methods.

Thank you for your comments, and we agree with you. Our description was obscure regarding our previous experiments. Our preliminary study was performed using only ICR strain mice and using the conditions of heat exposure described in our publication (Miyamoto et al. 2021), but we did not compare them with C57/BL6 mice at the same time. We rewrote this section to make it clearer.

  1. Line 38 – there has been a shift towards using the term “heat-related illness” instead of heat stroke within some medical field (See Yamamoto et al, 2015 and 2018 - http://www.mdpi.com/1660-4601/12/9/11770 and http://www.mdpi.com/1660-4601/15/9/1962), recognising that heat stroke is the most severe form of a continuous spectrum of heat induced illnesses. Consider re-phrasing?

Thank you very much. We have done so.

  1. Line 45 – please reference this temperature. For example Hausmann et al 2018 (http://link.springer.com/10.1007/s11606-018-4610-8) recently reviewed human normal temperatures and proposed different values.

We referred to the manuscript mentioned and changed the body temperature according to the reference.

  1. Line 49 – should “diffusing and vapouring” be “diffusion and evapouration”?

Line 70 – remove “basically”

Figure 1 – ensure legend is below figure.

Line 153 – please rephase “symptoms”, perhaps consequences? Symptoms are subjective, reported experiences of human patients so the term is not appropriate for an animal model.

Line 187 – “insignificant” consider changing to no significant differences.

Line 281 – “deteriorating role under heat stress” consider instead “had a detrimental effect under heat stress”.

Line 295 – consider rephrasing: “no significant differences in thermoregulation were present in normal non-stress…”

Line 305 – please remove bold

Thank you very much for your comments to improve our manuscript.  We modified these items accordingly.

  1. Line 80 – this may be a journal specific presentation style, but results are not normally presenting alongside the aims? Please check.

We removed the results from the Introduction. Thank you.

  1. Line 88 – as noted above, please clarify if this preliminary study was part of this research project or the cited work? “were weak against heat stress” consider rephasing to “were more susceptible to heat stress”.

Thank you for your comments and we agree with you. As you descibed above, we rephrased.

  1. Line 111 – please specify the groups. Also, did you perform any sample size estimations to determine the group size? Given the welfare impact on the mice used, it would be good to know that the minimum number possible were used in line with good ethical practice.

In the present model, the number of animals was an essential factor in inducing heat stress stably, and 10 animals (5 wild-type and 5 PACAP KO mice) were required in an experiment in the case of ICR mice. Moreover, before starting this experiment we expected that PACAP KO mice would be more vulnerable to heat exposure. We had difficulty believing the results because they were the opposite of what we had expected. Therefore, we performed the experiment again to confirm the reproducibility. We did not perform any sample size estimations. However, we think that the number of animals in the present study was the minimum necessary.

  1. Lines 193-200 – some phrasing in this section is unclear, e.g. “they were significant between -100 and -200”, do you mean there were significant differences in temperature, or in the rate of temperature change? Please clarify.

We agree with you. We revised the expressions.

  1. Line 213 – consider rephrasing “suppressed”, as this implies there is a mechanisms actively suppressing temperature increase, whereas you appear to report that instead the KO mice instead were protected from mechanisms that contributed to temperature increase? So perhaps “increases in the cT during exposure to heat stress were less pronounced in the KO mice”?

We changed the descriptions.

  1. Line 280 – “were better blood count” consider instead “had a higher survival rate against heat exposure than wild-type mice with fewer haematological and serum biochemical changes, including reduced evidence of hepatic damage”.

We have changed it.

  1. Lines 283 – 292 – consider expanding this paragraph to clarify the links between the main statements, at the moment the shorter standalone statements do not appear to be related?

We revised the discussion.

  1. Line 299 – “in this study” do you mean in the present study, or do you mean in study reference 36? Please clarify.

We changed this to clarify.

  1. Line 520 – please clarify that the wild type mice were ICR mice, at the moment this statement is misleading, as this heatstroke model used a lower ambient temp compared to the previous C57/BL6 mice.

So far, we do not clearly know the reasons why ICR mice were more vulnerable to heat stress than the C57/BL6 mice. However, we know a crowded condition in the chamber is very important to induce heat related illness. When we determined the conditions for heat exposure in C57/BL6 mice, we became aware of the importance of the animal number to produce a stable model. Therefore, we have always used a fixed number of animals for heat stress exposure. For example, we feel the heat in a crowded train even if the ambient temperature was the same as an uncrowded train.

The ICR mice have a larger body size than that of C57/BL6 mice. Therefore, we think that under same conditions and animal numbers, the situation became more severe for the ICR mice. Of course, we have not subjected both strains of animals to the heat stress test at the same time. Moreover, we have not compared serum parameters, tissue damage, the level of PACAP, and/or other factors. Therefore, we cannot eliminate the possible influence of other factors. Thank you very much for your comments. We will study the deteriorating and ameliorating factors to demonstrate the mechanism of heat related illness.

  1. Line 542 – “not applicable” is not appropriate, either the data should be made available in an open access formal, or this section to detail how interested parties can obtain the data, e.g. who to contact.

Thank you very much. We added the person to contact in the Data Availability Statement.

Reviewer 2 Report

Pituitary adenylate cyclase-activating polypeptide (PACAP) is a kind of canonical neuropeptide with multiple biological roles. Although the functions of PACAP have been reported in many disease studies, the present study clarified the possible roles of the neuropeptide in the process of heat stroke by PACAP knockout mice as an animal model. It provides a good tool and idea for heat stroke research.

The work and results are interesting, while major questions present as followings:

1.        As an essential material in this study, the results of PACAP gene knockout needed to be displayed. It is suggested to attach the results of Q-PCR or Western blotting of corresponding genes in PACAP knockout mice in the supplement.

2.        Whether the evaluation of the heatstroke model in this article has corresponding clinical standards? In the experience of heat exposure, there was no significant trend change in serum biochemical parameters and pathological changes between wild-type and PACAP KO mice. Although the core body temperature (cT) increased slower in PACAP KO mice, and expression of the c-Fos gene showed a difference, the conclusion acquired from these results lacks scientific evidence if there are no clear clinical quantitative indicators.

3.        It is suggested to consider the difference in the tolerance of mice and humans to the high ambient temperature or relative humidity environment.

Author Response

Response to Reviewer #2

Thank you very much for providing positive and important comments to improve our manuscript. We read your comments repeatedly and revised our manuscript accordingly. We also responded to the reviewer’s comments as follows:

  1. As an essential material in this study, the results of PACAP gene knockout needed to be displayed. It is suggested to attach the results of Q-PCR or Western blotting of corresponding genes in PACAP knockout mice in the supplement.

We added the supplementary method and results to determine the representative data for genotyping.

  1. Whether the evaluation of the heatstroke model in this article has corresponding clinical standards? In the experience of heat exposure, there was no significant trend change in serum biochemical parameters and pathological changes between wild-type and PACAP KO mice. Although the core body temperature (cT) increased slower in PACAP KO mice, and expression of the c-Fos gene showed a difference, the conclusion acquired from these results lacks scientific evidence if there are no clear clinical quantitative indicators.

These are very important comments. In most heat stress models, animals were subjected to very high temperatures without controlling relative humidity, and the models were lethal within a day. However, in human cases, many patients of heat-related illness survive, and some patients continue to suffer afterwards. To determine the reasons and develop new therapeutic strategies, we developed a WBGT-based (considering both temperature and humidity) heat stroke model (Miyamoto et al 2021). The animals in our model exhibited similar symptoms histologically and in biochemical parameters as human patients. Moreover, these damages were suppressed by rehydration, which is also recommended to prevent human heat-related illness.

Please review our results for serum biochemical parameters and pathological changes after heat exposure. Before heat exposure, neither group of animals showed any significant differences in any biochemical factors. However, after heat exposure, many biochemical factors were significantly different between the groups, and the parameters indicate dehydration, hepatic, and renal failures, along with other failures. The pathological changes also indicated hepatic damage even immediately after heat exposure. We think that other tissue damage progresses afterwards, because damage was detected in the liver, kidneys, and intestines in the C57/BL6 mice at 24 hours after heat exposure (Miyamoto et al, 2021). Of course, we must study further to produce a reliable comparison with human heat-related illness.

  1. It is suggested to consider the difference in the tolerance of mice and humans to the high ambient temperature or relative humidity environment.

Because we have not directly compared the two species, we cannot give an exact answer. However, humans and mice have different sweat gland distributions and heat vaporing systems. The tolerance for WBGT (a high ambient temperature or relative humidity in the environment) is different. However, we think that the progression of damage and bodily responses after heat stress may be similar to each other.

Reviewer 3 Report

The manuscript investigates the role of PACAP in heat stroke induced injury. The authors have attempted to propose the role of PACAP in protection from heat injury through non-shivering pathway.

The topic is well introduced. Results are well written but there are some missing details mentioned below. Discussion is bit long and misses the flow. Overall writing has several language mistakes and sentence restructuring is required. 

Some specific edits-

Page2 line 54- “through lipolysis of” change to through lipolysis by

Page 8 line 215-217 -Sentence needs modification

Page 10- Figure 6C is missing from the page

Page 12- sentence 290- remove bracket after DMH

Page 12- sentence 294-295- Sentence needs modification

Page-18- Sentence 513- correct spelling of statistical analysis. Currently says- statical.

Author Response

Response to Reviewer #3

Thank you very much for providing positive and important comments to improve our manuscript. We read your comments repeatedly and revised our manuscript. We also responded to the reviewer’ comments as follow:

  1. The topic is well introduced. Results are well written but there are some missing details mentioned below. Discussion is bit long and misses the flow. Overall writing has several language mistakes and sentence restructuring is required.

Thank you for your suggestion. We revised the discussion to be shorter and had the manuscript reviewed again by an English editing company.

  1. Page2 line 54- “through lipolysis of” change to through lipolysis by

Page 8 line 215-217 -Sentence needs modification

Page 10- Figure 6C is missing from the page

Page 12- sentence 290- remove bracket after DMH

Page 12- sentence 294-295- Sentence needs modification

Page-18- Sentence 513- correct spelling of statistical analysis. Currently says- statical.

Thank you. We revised all of these items.

Round 2

Reviewer 2 Report

The manuscript is acceptable.